# Endogenous Biological Drivers in Diabetic Lower Limb Wounds Recurrence: Hypothetical Reflections

**DOI:** 10.3390/ijms241210170

**Published:** 2023-06-15

**Authors:** Jorge Berlanga-Acosta, Ariana Garcia-Ojalvo, Gerardo Guillen-Nieto, Marta Ayala-Avila

**Affiliations:** Tissue Repair, Wound Healing and Cytoprotection Research Group, Biomedical Research Direction, Center for Genetic Engineering and Biotechnology, Playa, Havana 10600, Cuba; ariana.garcia@cigb.edu.cu (A.G.-O.); gerardo.guillen@cigb.edu.cu (G.G.-N.); marta.ayala@cigb.edu.cu (M.A.-A.)

**Keywords:** diabetic foot ulcer, ulcer remission, ulcer recurrence, ulcer relapse, diabetic complications, metabolic memory

## Abstract

An impaired healing response underlies diabetic foot wound chronicity, frequently translating to amputation, disability, and mortality. Diabetics suffer from underappreciated episodes of post-epithelization ulcer recurrence. Recurrence epidemiological data are alarmingly high, so the ulcer is considered in “remission” and not healed from the time it remains epithelialized. Recurrence may result from the combined effects of behavioral and endogenous biological factors. Although the damaging role of behavioral, clinical predisposing factors is undebatable, it still remains elusive in the identification of endogenous biological culprits that may prime the residual scar tissue for recurrence. Furthermore, the event of ulcer recurrence still waits for the identification of a molecular predictor. We propose that ulcer recurrence is deeply impinged by chronic hyperglycemia and its downstream biological effectors, which originate epigenetic drivers that enforce abnormal pathologic phenotypes to dermal fibroblasts and keratinocytes as memory cells. Hyperglycemia-derived cytotoxic reactants accumulate and modify dermal proteins, reduce scar tissue mechanical tolerance, and disrupt fibroblast-secretory activity. Accordingly, the combination of epigenetic and local and systemic cytotoxic signalers induce the onset of “at-risk phenotypes” such as premature skin cell aging, dysmetabolism, inflammatory, pro-degradative, and oxidative programs that may ultimately converge to scar cell demise. Post-epithelialization recurrence rate data are missing in clinical studies of reputed ulcer healing therapies during follow-up periods. Intra-ulcer infiltration of epidermal growth factor exhibits the most consistent remission data with the lowest recurrences during 12-month follow-up. Recurrence data should be regarded as a valuable clinical endpoint during the investigational period for each emergent healing candidate.

## 1. Introduction

The prevalence of both type 1 and type 2 diabetes (T1DM and T2DM) and the associated complications are globally increasing [1,2,3]. For T2DM, its incidence has progressively escalated, approaching a pandemic condition that accounts for 90% to 95% of the diabetic population [4].

Diabetes-affected patients have a significantly shorter life expectancy than non-diabetic individuals [5,6]. Accordingly, the seminal revolution brought about by insulin treatment did not translate into a significant reduction in chronic complications that preside morbidity and mortality [7].

Mounting evidence supports the existence of a diabetes metabolic memory as a proximal trigger in the perpetuation of multi-organ complications [8], including the torpid healing response [9]. Diabetic foot ulceration (DFU) is one of the most frightening diabetic complications, leading to amputation, disability, social exclusion, and early mortality [10]. The lifetime incidence of foot ulcers has been estimated to reach up to 34% of diabetic subjects, contributing to 80% of all non-traumatic lower extremities amputations around the world [11,12].

Although the onset of DFU is associated with predisposing factors such as diabetes evolution data, poor glycemic control, smoking habit, high blood pressure [13], peripheral neuropathy, limb ischemia, and cutaneous frailty [14], diabetic patients are affected by an intrinsic healing impairment that spans from the oral cavity mucosa, to lower extremities peripheral soft tissues [15,16,17]. DFU chronicity phenotype [18] seems to be driven by precocious senescence, proliferative arrest, and apoptosis as distal effectors of an abnormal hyperglycemia-associated epigenetic code [19,20].

In addition to the impaired healing response, a parallel conundrum in diabetic ulcer pathology is the high rate of recurrence after the primary re-epithelialization [21,22]. In line with this fact, a classic report reveals that roughly 40% of diabetic patients have a recurrence within 1 year after ulcer healing [23,24]. Additionally, an underappreciated risk of DFU recidivism is its ability to “metastasize” at anatomical niches away from the primary occurrence, frequently leading to a subsequent amputation [25].

Ulcer recurrence investigation has mostly focused on the identification and validation of predisposing clinical factors such as poor glycemic control, previous ulcers, wound healing time, local skin damage, bone deformities, neuropathy, ischemia, diabetes duration, end-stage renal disease, toxic habits, and scar tissue mechanical stress [24,26,27]. Although the deleterious role of behavioral factors is unquestionable for both DFU onset [13] and the episode of recurrence [23,28], we assume that there are endogenous biological signalers that prime the residual scar tissue. Studies addressed identifying molecular pro-recurrence predictors are a contemporary need. We consequently hypothesize on the potential molecular and cellular drivers of local or systemic origin that may concertedly cooperate behind ulcer recurrence. Having reviewed the literature, we may assume that: (1) Endogenous deterrent factors underpinning the diabetic healing deficit are likely the same that cooperate for ulcer relapse. (2) These endogenous factors may be represented by soluble circulating signalers as by in-scar anchored dysfunctional cells that secrete “pathological messages”. (3) These factors are a consequence of hyperglycemia, whereas some of them are glucose-derived chemicals. (4) Both wound chronification and recurrence are influenced by abnormal cell physiology on the basis of an epigenetic code resulting from the interaction of glucose and its derivatives with expression/transcription regulatory factors. (4) Numerous published clinical trials on innovative treatments for the healing of DFU omit mention of recurrence rates during post-epithelialization follow-up. Intervention with locally infiltrated epidermal growth factor (EGF) appears as a promising tool to achieve prolonged remission times.

The information analyzed for this work was retrieved from Pubmed and Google Scholar databases, restricted to the English language, with no date limitation.

### 1.1. Potential Epigenetic Drivers in Ulcer Recurrence

Scar tissue is vulnerable to recurrence, an event that is theoretically propelled by abnormal cellular physiology with an underlying epigenetic program and a chemically-modified extracellular matrix by hyperglycemia-associated toxic products [19]. Thus, ulcer relapse is similar to impaired diabetic healing, and both are clinical consequences of poor glycemic control, highlighting the potential pathogenic participation of endogenous cytotoxic-recurrence primers [22]. Blood glucose levels and hyperglycemia-derived products constitute an environmental factor that impacts the plasticity of epigenetic mechanisms, thereby modifying the whole transcriptome of skin cells [29]. Compelling data confirm the influential role of epigenetic regulation in diabetic complications. Most relevant cellular processes, such as plasticity, adaptability, differentiation, replication, and memory, may be governed by epigenetic mechanisms. Accordingly, epigenetics is the tool by which the cell regulates gene expression without alteration of the underlying DNA sequence [20]. Thus, cellular gene expression and phenotypic traits are coordinated by an epigenetic program, largely impinged by environmental factors such as glucose [30]. Interestingly, simple chemical modifications of nucleic acids and histone proteins as DNA methylation, histone modifications, chromatin remodeling, and expression of non-coding RNA (ncRNA), are the main driving tools in epigenetic regulation [31]. Although changes in DNA methylation patterns and other epigenetic changes are considered reversible modifications, they can be mitotically and meiotically inherited and, therefore, intra and trans-generationally transmitted [32]. Of note, glycemia can impact cell homeostasis at both genetic and epigenetic levels. A primary multi-organ effect originating from deregulated glycemia consists in disrupting three major biological functions: DNA expression, RNA transcription, and protein translation. Studies have indicated that hyperglycemia may increase DNA mutations, DNA breaks, genomic instability, and particularly epigenetic dysregulation [33,34]. Accordingly, hyperglycemia and a constellation of downstream factors transform the cellular native epigenetic architecture, rendering a de novo code that largely alters cellular physiology [35]. Although the exact mechanisms underlying the involvement of DNA methylation in T2DM pathogenesis remain unclear [32], an association exists among hyperglycemia, the pattern of DNA methylation changes, diabetes complications unset, and their clinical progression [36].

Abnormal DNA methylation predisposes one to diabetes susceptibility genes [37] and perpetuates the diabetic phenotype in ulcer-derived fibroblasts, which is independent of the number of culture passages and the presence of a normal glucose concentration in the culture medium. The discovery of this epigenotype supports the role of epigenetics in the persistent diabetic phenotypic behavior of the ulcer fibroblast population [38]. Another form of epigenetic modification, histone methylation, has been shown to correlate with glycemia levels and, in turn, with the formation of advanced glycation end products (AGEs) and the ensuing hyperglycemia-related inflammation, oxidative stress, and apoptosis [39]. Both DNA methylation and demethylation are abnormal in diabetic wound healing under high-glucose conditions, resulting in the disruption of the normal progression of the healing phases [9]. Convincing evidence for the role of an aberrant DNA methylation pattern in diabetic wounds was discovered by Babu and co-workers, showing a global hypomethylation in promoters of pro-inflammatory class 1 phenotype macrophages (M1) accounting for a prolonged inflammation and reduced angiogenesis [40]. Other studies have indicated that complete or partial methylation of the TLR2 gene promoter is associated with its functional downregulation, leading to poor wound healing in diabetic patients. Of note, TLR2 is a family member of the Toll-like receptors with an active role in pathogen recognition and activation of innate immunity, modulating endothelial cell migration, angiogenesis, and wound healing [41]. Contributing to the shedding of light on the mechanistic basis of the poor angiogenic response in diabetic wound healing is the observation that endothelial cells exposed to transient hyperglycemia exhibit enhanced methylation of the angipoietin-1 promoter with the subsequent downregulation of its expression. Moreover, transient hyperglycemia concomitantly resulted in sustained activation of nuclear factor-κB (NF-κB) and ensued endothelial dysfunction [42]. Other findings converge to indicate that inhibiting the activity of methylation enzymes on the promoter of VEGF receptor Flt- contributes to the enhancement and acquisition of endothelial cells phenotype of mesenchymal stem cells [43]. In a similar manner, AGEs, as a biochemical hallmark of diabetes and a crucial ingredient within the damage cascade, can transform the transcriptional landscape rendering an abnormal transcriptome [44,45] upon sticking to histones, thereby modifying chromatin structure and its interaction capabilities. These mechanisms are graphically summarized in Figure 1.

It is worth indicating that aberrant DNA methylation is likely a trait associated with wound chronicity and not exclusive to diabetic ulcers. An elegant study based on unbiased whole-genome methylome of the edges of chronic wounds concluded that DNA hypermethylation is more common than DNA hypomethylation at chronic wound margins; (ii) that results in downregulation of DNA hypermethylated genes, inhibiting epithelial-mesenchymal transition and impairing wound epithelialization; and that (iii) correction of DNA hypermethylation is an effective mechanism in improving wound closure [46].

Chronic high glucose level is the primary precursor of AGEs formation and accumulation in tissues, which is linked to ROS and LPP formation. Hyperglycemia, AGEs, and ROS have a direct cytotoxic effect which includes damaging cellular and mitochondrial DNA, and respiratory activity, which further amplifies ROS production. The above biochemical reactants introduce novel epigenetic imprinting through a DNA aberrant methylation pattern, abnormal transcription, and expression profiles. This foundation of metabolic memory perpetuates the diabetic phenotype in fibroblasts and keratinocytes, which includes inflammatory and senescence activators in proliferative arrest.

As mentioned above, hyperglycemia also modifies the transcriptional profile by affecting mRNA, transcription factors, the production and extracellular release of microRNAs (miRNA), and long non-coding RNA (lncRNA) [47]. El-Osta’s findings inaugurated the contention that exposure to transient hyperglycemia was sufficient to reprogram cells’ native epigenetic program and that these modifications largely persisted even beyond the normalization of glucose levels [48]. In other words, the acute exposure of cells to high glucose stress translates into chronic consequences that involve the onset of lasting pro-inflammatory and oxidative programs upon an altered epigenetic code. This concept was further validated in cultured human skin primary fibroblast [29] and vascular cells while showing that high glucose concentrations introduced significant transcriptomic modifications in genes controlling multiple pathways, all involved in wound healing events, including angiogenesis [49].

MicroRNAs (miRNA) and non-coding RNAs, in general, are involved in an extensive array of cellular functions, representing, as just mentioned, an additional layer of epigenetic control in cell physiology [33]. The deficiency of miRNA biogenesis has revealed their biological significance in the skin healing process [50]. Interestingly, there is a substantial specificity and differentiation of miRNA profiles for the specific type of diabetes and the evolving complications, which has encouraged its application as diagnosis and prediction biomarkers [50,51]. A large number of microRNAs have been implicated in pathological diabetic healing [50,51,52,53,54,55]. A recent study identified the enhanced expression of miR-155 in the peripheral blood of T2DM patients as a potential predictor for the onset of DFU [56], whereas the expression level of miR-203 in patients with DFU positively correlated with the severity of the damage [57]. A high expression level of miR-34c positively correlated with the amputation rates while negatively with the healing response and was also identified as an independent risk factor for ulceration and osteomyelitis [58]. Likewise, long non-coding RNAs (lncRNAs) are an important epigenetic regulator at the level of histone methylation and gene transcription [47,59]. lncRNAs play a substantial role in diabetic wound healing, encompassing infiltrating macrophage polarization control and keratinocytes proliferation and migration. Differential expression of lncRNAs in diabetic patients shows their involvement in impaired diabetic healing and their putative role as biomarkers for diabetes-mediated damages [59,60,61]. Although miRNAs and lncRNAs are included in the broad collection of hyperglycemic stress-related epigenetic derangements [62], it still remains elusive if some of the non-coding RNA forms are associated with scar tissue relapse and could therefore be used as predictive biomarkers.

Conclusively, genetic and epigenetic cell resources are impacted by chronic hyperglycemia establishing an abnormal epigenetic program, largely based on aberrant DNA methylation, that predisposes to the onset and perpetuation of diabetic traits such as cellular senescence, proliferative quiescence, inflammation, oxidative imbalance, and apoptosis [49]. Thus, scar tissue integrity is intrinsically jeopardized by these underlying silent primers, which may lead to tissue death as the ultimate event in recurrence, validating the concept of ulcer remission instead of ulcer healing.

### 1.2. Dermal Matrix, Fibroblasts, and Keratinocytes in Ulcer Recurrence

Diabetes disrupts skin structure and physiology. Chronic and irregular courses of glucose and glucoxidation-derived products undermine skin cell physiology and progressively intoxicate the dermal matrix by the accumulation of AGEs, nitrosilation products, and free radicals byproducts [63,64]. Diabetics’ intact skin shows reduced biomechanical resilience and stress tolerance, diminished elasticity, increased stiffness, and reduction in collagen and elastin content, altogether contributing to healing impairment [51,65] and predisposing to scar tissue and epithelial coverage fractures. The above-described dermal anomalies are largely associated with the accumulation of AGEs crosslinked with long half-life proteins such as collagen, which irreversibly and progressively affects skin matrix mechanics, induces premature aging, and impairs critical healing events such as angiogenesis, fibroblasts attachment, and myofibroblasts-induced contraction [66,67]. Both AGEs and oxidative stress have a direct cytotoxic effect on skin fibroblast physiology, are instrumental ingredients of the ulcer cytopathic milieu, and ultimately contribute to molding the diabetes epigenetic map (Figure 2) [68,69]. It is, therefore, inferable that scar tissue cells are born embedded within a milieu permeated by cytotoxic products.

Hyperglycemia and its metabolic-associated derivatives, such as AGEs and ROS, accumulate in the dermal collagen and alter the chemical and physical properties of the skin, becoming vulnerable and frail. These are also cytotoxic products for dermal fibroblast physiology and survival. Hyperglycemia and the downstream cytotoxic products damage the native DNA methylation pattern, subsequently modifying gene transcription and promoting a diabetic phenotype of keratinocytes and fibroblasts by a de novo re-written epigenetic code.

Another pathogenic ingredient predisposing to ulcer recurrence is the unusual premature skin cells’ aging brought by hyperglycemia and its derivatives [70,71,72]. Diabetes is a mitochondrial-related disease, and not surprisingly, mitochondrial dysfunction is considered a primary trigger of skin aging and other phenotypic manifestations, such as impaired healing [72]. A variety of mitochondrial and other diabetes disorders include hyperinflammation, high proteolytic activity, defective oxidative phosphorylation, local hypometabolism, excessive ROS generation, and accumulation of AGEs found in skin fibroblasts and keratinocytes [73,74], which translate into healing impairment [75]. We deem that the persistence of these factors and/or their epigenetic signature may contribute to ulcer recurrence. In line with this notion, magnetic resonance studies confirmed that edema and hypometabolism of lipids and amino acids persist during remission time rendering scar tissue vulnerable [76]. Decisively, diabetes biochemical derangements are a major and direct skin-aging driving factor, causing cellular dysfunction and dermal proteins denaturation and decay [77,78].

Skin fibroblasts are a functionally heterogeneous mesenchymal cell population with a central role in wound repair. These are sensitive cells with a large reserve of plasticity and reprogramming before external clues, which may alter their biological behavior and, ultimately, wound-healing fate [79]. Most importantly, fibroblasts are cells endowed with the ability to retain a memory from their positional location, mechanical and inflammatory environments, and especially a metabolic memory. Accordingly, fibroblasts can sense intracellular and extracellular metabolic changes in their microenvironment and consequently orchestrate a long-lasting phenotypic response [80]. Thus, the society of wound fibroblasts may modify the course of the healing process as the long-term fate of the residual scar [38,69]. Short-term exposure of cultured, healthy, non-diabetic donor dermal fibroblasts to a high glucose burden hampers proliferation, anabolism, and migration signaling pathways and orchestrates senescence [81], thus mirroring the phenotypic pattern detected in diabetic fibroblasts explanted from foot ulcers [69,82]. Hyperglycemia causes apoptosis of dermal fibroblasts, reduces collagen expression, and upregulates RELA/p65 expression, which implicates the onset of pro-inflammatory and pro-degradative profiles [63,83,84]. An increased repertoire of inflammatory biomarkers is associated with non-healing DFU, whereas an elevated neutrophil-to-lymphocyte ratio (NLR) shows a positive correlation with increased risk of amputation and ulcer septic complications [85]. The presence of endogenous skin-damage predisposing markers was identified by single-cell transcriptome studies of DFU specimens, in which multiple fibroblast cell clusters showed an increased inflammation pattern, changes that were likewise detected in areas of intact skin of diabetic subjects [86]. Thus, scar tissue aftermath may depend on fibroblast metabolism, its secretory capability of extracellular proteins, and its control over the quality and duration of the inflammatory reaction [75].

In addition to their critical physiology, keratinocytes are veteran sentinel cells that initiate the healing cascade after epidermal integrity is disrupted [87]. Similar to dermal fibroblasts, keratinocytes are also memory cells [88,89] whose “response to wounding” is seriously affected by high-glucose stress [90]. Hyperglycemia introduces alterations of keratinocytes metabolism, adhesion, migration, proliferation, and differentiation [91,92], all having an epigenetic fundamental through abnormal changes in DNA methylation [93,94,95]. Similarly, the onset of a senescent phenotype by epidermal cells may be a major molecular gear for ulcer recurrence. Hyperglycemia and its chemical derivatives shape an epigenetic landscape in which upregulation and post-translational modifications of p53, p21, and p16 contribute to keratinocyte senescence [19]. Hypothetically, significant for ulcer recurrence could be the active expression of matrix metalloproteinase-9 (MMP-9), a type IV collagenase expressed by keratinocytes at the wound’s leading edge, which may hinder re-epithelialization when upregulated by AGEs. MMP-9 levels are elevated by hyperglycemia and glycation products via the upregulation of ten-eleven translocation enzyme 2 (TET2) gene expression. TET2 expression is higher in epidermal cells of diabetic patients than in normal skin, which appeared to be a consequence of high levels of α-ketoglutarate. Of note, the levels of α-ketoglutarate correlate with local hypoxia, ischemia, and with poor glycemic control, exemplifying how the local environment and metabolism impact wound cells physiology via epigenetic mechanisms [9,96].

An intrinsic fragility of the epidermal layer may be a predisposing factor for ulcer recurrence. One of the hallmarks of diabetic wounds is the high rate of keratinocyte proliferation versus an unsuccessful differentiation platform. Studies in diabetic mice have demonstrated an abnormal skin differentiation program due to an underlying keratinocyte dysfunction. Human non-diabetic keratinocytes exposed to hyperglycemic stress and diabetic subjects-derived epidermal cells exhibit a common differentiation dysfunction mediated by overexpressed c-Myc, which blunts differentiation by activating the WNT/β-catenin pathway [97]. Previous observations had already indicated that activation of the β-catenin pathway and an enhanced expression of c-Myc, were implicated in the diabetic torpid re-epithelialization response by disrupting keratinocyte migration and differentiation [98].

We also deem that a successful reciprocal and dynamic communication between epidermal keratinocytes and dermal fibroblasts is mandatory to ensure scar tissue health and, accordingly, prevent wound recurrence. This notion is founded on the evidence of signaling crosstalk between these two major cell lineages. Keratinocytes nurture fibroblasts and myofibroblast activity via the paracrine secretion of vascular endothelial growth factor-A (VEGF-A), transforming growth factor-β1 (TGF-β1), and connective tissue growth factor (CTGF), whereas reciprocally fibroblast-derived TGF-β1 expression, is essential for keratinocyte physiology including migration [99]. High glucose levels and AGEs are known to disrupt this dermo-epidermal cells homeostatic circuit by reducing forkhead box O1 transcription factor (FOXO1) expression in keratinocytes, thus hindering keratinocytes’ ability to produce TGF-β1 (Figure 3) [99]. Altogether, these findings converge to highlight: (1) the impact of glycemic control in keratinocyte physiology, (2) the significance of keratinocyte epigenetic imprinting on scar tissue integrity, and (3) the biological significance of a healthy dermo-epidermal axis.

## 2. Ulcer Recurrences in the Clinical Arena

Although recurrence rate reports of diabetic foot ulcers appear to differ broadly in the current literature [100], statistical data are alarmingly high despite the variety of healing interventions and improved multidisciplinary management of the condition [21,101]. The literature on ulcer recurrence quite often misses data on the most specific recurrence time points and anatomical sites [21]; unfortunately, not all the clinical trials examining the healing efficacy of drugs, devices, or management approaches include information about recurrence rates over a reasonable follow-up period [102].

Although during the past 20 years, there has been an explosion in basic science-derived approaches that include wound dressings, living cell equivalents, and smart growth factor (GF) formulations for efficient local delivery; some of these candidates still need clinical validation, and others vanished along the way due to limited therapeutic impact [103]. The sequential discovery of GFs at the beginning of the 1960s and their biological capability to enhance cell proliferation and migration, and to circumvent cell cycle arrest, fueled their introduction to treating a variety of acute and chronic wounds and ever since progressed up to the development of nano-formulations [103,104]. Further studies in the early 1990s, however, suggested that the clinical use of GFs was somewhat precipitated, given that chronic wounds milieu impaired the response to GFs by a variety of biochemical and physical factors including in situ degradation, limited diffusion, and reduced receptors catalytic activity [105]. As for DFU, the recombinant PDGF-BB in its pharmaceutical presentation form (Regranex) has been the only FDA-approved GF since 1997 https://www.accessdata.fda.gov/drugsatfda_docs/label/1997/becaomj121697-lab.pdf, accessed on 15 May 2023).

Apligraf^®^ inaugurated the era of regenerative medicine in which lost or damaged human cells are replaced by engineered cellularized wound dressings. It was the first tissue-engineered composite skin equivalent indicated to treat chronic diabetic and venous ulcers. It consists of a bioscaffold matrix cultured with allogeneic male neonatal fibroblasts and keratinocytes [106,107]; subsequently, Dermagraft^®^ appeared, an allogeneic dermal replacement made from foreskin fibroblast cells combined with biodegradable mesh. Once implanted in the wounds, the cryopreserved fibroblasts acquire viability and create growth factors and extracellular matrix (ECM) components that foster the healing response [108]. These cell-based dressings for extensive burn injuries and chronic wounds, ranging from traditional and “second generation” bioengineered living skin equivalents to mesenchymal stem cell dressings, represent a shift in the wound healing paradigm [109,110].

As an innovative device, negative pressure wound therapy (NPWT), also known as vacuum-assisted closure (VAC), is a biophysical agent consisting of a mechanical unit attached to a dressing through a plastic tube which, when connected to a suction device, enables the creation of sub-atmospheric pressure at the site of a wound [111]. NPWTs mechanism of action appears to reside in pulling wound edges together to narrow the wound size, promoting granulation tissue formation on the wound bed for skin-grafting, enhancing microcirculation, decreasing edema, and removing infectious ingredients [112]. Nevertheless, a recent study has concluded that still, it remains unclear whether NPWT is more effective for the treatment of DFUs than any other dressing type [111].

Here we present the recurrence rates data of relevant clinical investigations (clinical trials, different meta-analyses, and a comprehensive review), comprising the three major groups of first-in-line products: recombinant proteins as EGF and PDGF-BB, cellular and/or tissue-based products, and devices (vacuum-assisted closure/negative pressure wound therapy). Of a total of 24 materials reviewed with these products, recurrence data were reported in only 10 (41.6%).

According to these collected data, post-epithelialization recurrence incidences are high, especially those observed in the nationwide phase III clinical trial in the USA for Regranex/Becaplermin [113]. Graftskin (Apligraf) exhibited the same incidence of ulcer recurrence as the control group, whereas the Dermagraft study does not include these data. Surprisingly, the incidence of recurrence is not described in any of the five studies reviewed (Table 1). Inversely, EGF treatment based on intralesional infiltration delivery appears to provide the longest remission times with the lowest recurrence rates (Table 1). A recent systematic review of randomized controlled trials investigating different recombinant GFs for the purpose of wound healing concludes that EGF is the most effective GF to enhance DFU healing [102]. A similar conclusion is drawn from another meta-analysis stratified by the types of administration route (intralesional injection and topical administration) in which six studies involving 530 patients were eligible for review [114]. It is likely that EGF is the most broadly studied GF in wound healing [103], and interestingly, its healing effects are far more notorious as the wounds are bigger [115], suggesting that EGF is endowed with a broad therapeutic window. In addition, it is likely that the success of the infiltrated EGF in prolonging scar tissue integrity may reside in its ability to positively impact the skin cell’s abnormal epigenetic program and/or reduce the society of senescent fibroblasts [19]. Definitively, the durability of ulcer-free time should be regarded as a primary endpoint of major benefit in clinical trials for developing products/treatments [116].

## 3. Concluding Remarks and Future Directions

Diabetes is likely the most convincing scenario to illustrate how a trivial exogenous factor, such as the level of blood glucose, may reshape the native epigenetic program and ultimately build a disease-pathologic memory. The existence of this hyperglycemia-mediated metabolic stress memory explains why prior hyperglycemic exposure is not “forgotten” with time and successive cell generations. On the other hand, diabetes is an exemplary disease in terms of the generation and the progressive accumulation of hyperglycemia-derived cytotoxic products, some of them being cumulative in tissues such as the skin. Accordingly, the conjunction of epigenetics and the consequent abnormal cellular behavior, along with the chronic cytotoxicity exerted by the spillover of AGEs and free radicals, are biological factors that impair the healing response and continuously jeopardize scar tissue homeostasis, stability, and viability. It is not surprising, therefore, that under this environment, societies of senescent and mitosis-refractory cells are found entrenched within the residual scar and the skin of diabetics in general. The fact, as stated by Armstrong and co-workers, that recurrence may globally affect up to 40% of the patients in the first year after re-epithelialization, in addition, to being distressing, may indicate that: (1) the alert sense implicit in the ulcer remission concept has not sensitized enough to patients and wound care providers, (2) glycemia control may remain insufficient in the post-healing period, (3) endogenous, biological drivers remain silently active deteriorating the homeostasis of scar tissue and cells, (4) not all the innovative treatments contemporarily accepted to enhance acute ulcer healing, translate in prolonged remissions and far less in definitive healing, (5) treatments to be considered as effective are called to promote scar tissue resilience and offer a reasonable remission time.

Here we have endeavored to dissect endogenous molecular drivers that may be implicated in the episode of ulcer recurrence after re-epithelialization. This seems to be an empty niche in translational medicine, whereas circulating or in-scar molecular culprits and, consequently, predictors remain to be identified. In the meantime, two major lessons with clinical implications may be drawn from this theoretical study: (1) strict glycemic control remains the mainstay in the prevention of diabetic complications, and (2) the constant alert sense of the concept of remission must be the everyday scope for the patient and health care provider. These could be the two major pillars to reduce post-closure recurrences.

We hold the argument that future innovations for chronic wounds and accordingly prevent ulcer relapses must ideally entail the ability to target the cell epigenetic core, which could erase the chronic hyperglycemic stress memory, reduce the burden of senescent cells, and impose “healthy” re-differentiation programs. Manipulating the diabetic metabolic epigenetic code may indefectibly assist in the control of all its chronic complications. Of note, however, this therapeutic dream line must be anticipated by the well-deserved glycemic control.

## Figures and Tables

**Figure 1 ijms-24-10170-f001:**
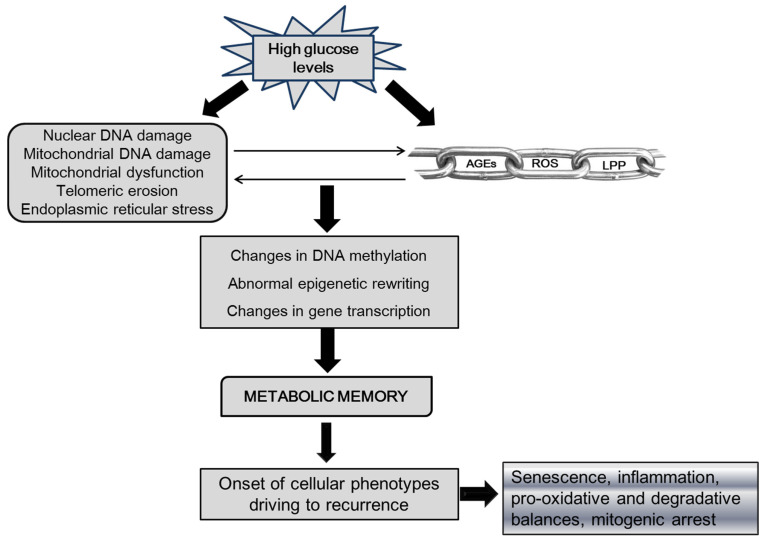
Theoretical epigenetic bases of recurrence. Legend: AGEs-advanced glycation end-products, ROS-reactive oxygen species, LPP-lipoperoxidation products.

**Figure 2 ijms-24-10170-f002:**
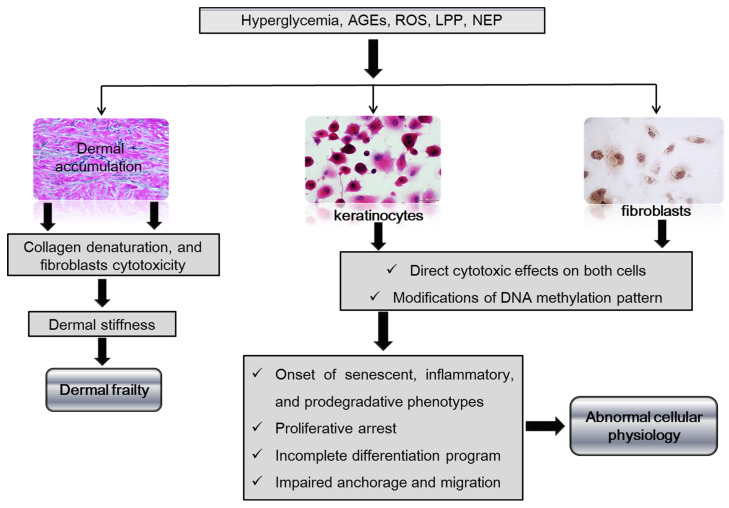
Impact of hyperglycemia and derived cytotoxic products on the dermal matrix and skin cells. Legend: AGEs-advanced glycation end-products, ROS-reactive oxygen species, LPP-lipoperoxidation products, NEP-nitrosylation end-products.

**Figure 3 ijms-24-10170-f003:**
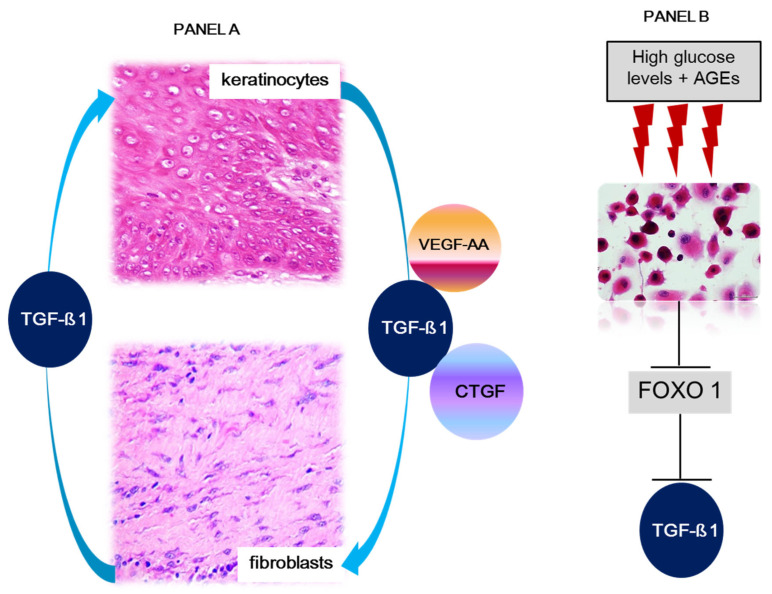
Remission duration may depend on dermo-epidermal axis homeostasis. Legend: VEGF-A, vascular endothelial growth factor-A, TGF-β1, transforming growth factor-β1, and CTGF, connective tissue growth factor. FOXO1, forkhead box 1. (**A**) Dermal fibroblasts and epidermal keratinocytes conserve dynamic and reciprocal crosstalk with the exchange of signalers for both cell populations’ homeostasis. Under physiological conditions, epidermal keratinocytes secrete growth factors involved in fibroblast physiology, including the synthesis, secretion, and turnover of dermal matrix proteins. This circuit includes the secretion by fibroblasts and utilization by the keratinocytes compartment of TGF-β1. In this context, this growth factor appears to participate in keratinocyte stability, migration, turnover, and differentiation. (**B**) In diabetic subjects, and hyperglycemia experimental settings, high glucose levels and AGEs result in significantly reduced TGF-β1 expression by keratinocytes. This event is due to FOXO1 molecular changes, which consequently reduces FOXO1’s ability to promote TGF-β1 expression.

**Table 1 ijms-24-10170-t001:** Post-healing recurrence rates reported in major clinical studies of biologics, cell and/or tissue-based products, and VAC.

Recombinant Epidermal Growth Factor (EGF)				
Reference	Major Outcome	Administration Route	Follow Up Period	Recurrences
Tsang, M.W. et al., 2003 [117]	20 of 21 diabetic foot ulcers healed with daily application of 0.04% (wt/wt) hEGF for 12 weeks	Topical	6 months	ND
Hong, J.P. et al., 2006 [118]	Topical treatment with EGF combined with advanced dressing may have positive effects in promoting healing	Topical	6 months	No recurrences were observed in EGF group
Park, K.H. et al., 2018 [119]	The phase III study supports the efficacy and safety of spray-applied EGF treatment for DFUs	Topical	ND	ND
Viswanathan, V. et al., 2006 [115]	In EGF group, 90% of ulcers healed in 15 weeks compared with 22 weeks in control group	Topical	ND	ND
Tuyet HL et al. 2009 [120]	Topical EGF spray has positive effects on healing of moderate-to-severe foot ulcers	Topical	ND	ND
Fernández-Montequin, J. et al., 2009 [121]	Locally infiltrated EGF at 75 µg enhanced granulation tissue growth and wound closure	Intralesional injection	12 months	No recurrences reported for EGF groups
Gomez-Villa, R. et al., 2014 [122]	Patients with DFU who received intralesional rhEGF application resulted in complete healing	Intralesional injection	ND	ND
Bartın, M. & Okut, G. 2022 [123]	Intralesional administration of EGF in T2DM can prevent amputations in DFU	Intralesional injection	6 months	Two cases in the group receiving EGF
Yera-Alos, I.B. et al., 2013 [124]	Post-marketing study including 1788 patients treated with intranuclear-injected EGF. Re-epithelization was documented in 61% of the 1659 followed-up cases	Intralesional injection	14 months	5%/year
López-Saura, P.A. et al., 2013 [125]	Intralesional use of EGF for high-grade DFU in more than 2000 subjects. It confirms the 75% probability of complete granulation response, 61% healing, and a 16% absolute, and 71% relative reduction in amputation risk	Intralesional injection	12 months	The frequency of relapseswas significantly lower (*p* < 0.001) in patients that received rhEGF:
Kahraman, M. et al., 2019 [126]	Study aimed to investigate the long-term outcomes after intralesional epidermal growth factor injections in the treatment of 34 diabetic patients with foot ulcers.	Intralesional injection	60 months	Of 29 patients involved in the 5-year follow-up, 27 were ulcer free
**Regranex or Becaplermin (rh-PDGF-BB)**				
**Reference**	**Major Outcome**	**Administration Route**	**Follow Up Period**	**Recurrences**
Embil, J.M. et al., 2000 [127]	Confirms the efficacy and safety of becaplermin gel for the treatment of lower extremity diabetic ulcers	Topical	6 months	21% of recurrence in Becaplermin-treated patients
Smiell, J.M. et al., 1999 [128]	Becaplermin gel at a dose of 100 μg/g once daily is effective in patients with lower extremity diabetic ulcers	Topical	3 months	ND
Wieman, T.J. et al., 1998 [113]	Becaplermin gel 100 μg/g significantly increased the incidence of complete wound closure	Topical	3 months	Ulcer recurrence incidence was ≈30% in all the groups
Ma, C. et al., 2015 [129]	Topical platelet-derived growth factor does not appear to significantly improve healing of Wagner grade I diabetic foot ulcers	Topical	6 months	No difference was observed between groups in recurrence
**Cellular and Tissue-Based Products**				
**Reference**	**Major Outcome**	**Administration Route**	**Follow Up Period**	**Recurrences**
Veves, A. et al., 2001 [130]	At the 12-week follow-up visit, 63 (56%) Graftskin-treated patients achieved complete wound healing compared with 36 (38%) in the control group (*p* = 0.0042).	Topical–bioengineered skin substitutes	6 months	The incidence of ulcer recurrence was similar for Graftskin and control groups
Marston, W.A. et al., 2003 [131]	Patients experienced a significant clinical benefit when treated with Dermagraft versus patients treated with conventional therapy alone.	Topical–bioengineered skin substitutes	ND	ND
Zelen, C.M. et al., 2016 [132]	EpiFix^®^ (dehydrated human amnion/chorion membrane) is superior to standard wound care SWC and Apligraf^®^, in achieving complete wound closure within 4–6 weeks.	Topical–bioengineered skin substitutes	ND	ND
Zelen, C.M. et al., 2014 [133]	DFU healed with use of dehydrated human amnion/chorion membrane (EpiFix) in 18 available subjects with healed DFU. Wound median size of 1.7 cm^2^.	Topical–bioengineered skin substitutes	9–12 months	17 wounds remained healed
**Vacuum Assisted Closure (VAC)/Negative Pressure Wound Therapy (NPWT)**				
**Reference**	**Major Outcome**	**Administration Route**	**Follow Up Period**	**Recurrences**
Blume, P. et al., 2008 [134]	A greater proportion of foot ulcers achieved complete ulcer closure with NPWT (73 of 169, 43.2%) than with advanced moist therapy within the 112-day active treatment	Topical–sub-atmospheric pressure over the wound area	ND	ND
Armstrong, D.G. et al., 2005 [135]	More patients healed in the NPWT group than in the control group (43 [56%] vs. 33 [39%], *p* = 0·040)	Topical–sub-atmospheric pressure over the wound area	ND	ND
Zhang, J. et al., 2014 [136]	Meta-analysis concludes that NPWT appears to be more effective for diabetic foot ulcers compared with non–negative-pressure wound therapy and has a similar safety profile.	Topical–sub-atmospheric pressure over the wound area	ND	ND
Liu, S. et al., 2017 [137]	Meta-analysis concludes that NPWT is efficacious, safe, and cost-effective in treating DFUs.	Topical–sub-atmospheric pressure over the wound area	ND	ND
Meloni, M. et al., 2015 [138]	A comprehensive review. DFUs managed with NPWT benefit from a significant reduction in the ulcer size and often a shorter treatment in comparison with ulcers treated with traditional gauze dressing	Topical–sub-atmospheric pressure over the wound area	ND	ND

ND—not defined.

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
