# Peer review of "Endogenous Biological Drivers in Diabetic Lower Limb Wounds Recurrence: Hypothetical Reflections"

_ijms, 2023, doi:10.3390/ijms241210170_

Round 1

Reviewer 1 Report

This review is focused on endogenous drivers in diabetic lower limb wounds recurrence. Despite, the topic is very interesting and original and it is very well written a major revision is needed to make the paper suitable for publication in this Journal.

Major comments:

1     For the relevance of the scientific Journal, the authors should better elucidate the molecular/epigenetic mechanisms that underlie the diabetic lower limb wounds recurrence after hospitalization.

2     According to the authors, is there a correlation between epigenetic modifications and biochemical markers in predicting diabetic lower limb wounds recurrence?

3     In the “conclusion” section, the authors should better highlight what could be the clinical and therapeutic strategies to reduce the risk of recurrence.

Minor comments:

1     Please use always T2DM as acronym for type 2 diabetes and not “T2-DM”

2     Please format Table 1 better

3     Please include the legend in the figure captions

4     It is recommended to create a final figure that illustrates the focal points of the review.

5      It would be useful to add a new reference about the impact of glucose control, diabetes-related complications on the risk of diabetic foot ulcers (DFUs) and DFU complications in patients with Type 2 Diabetes. For better support your review, the following references should be recommended: “Risk factors for diabetic foot ulcers: an Albanian retrospective study of inpatients with type 2 diabetes” (PMID 35113432). Identifying patterns in multiple biomarkers to diagnose diabetic foot using an explainable genetic programming-based approach (doi.org/10.1016/j.future.2022.10.019)

Author Response

Please find the attchment

Reviewer 2 Report

The review paper of Berlanga-Acosta J et al presents an up-to-date about the pathogenesis of diabetic foot wound recurrence and the new therapies proposed in the last 20 years.

This topic is very interesting but the pathogenesis should be better presented, and here, there could be some points:

1.      The concept of epigenesis should be defined and well explained before using it 20 times in 5 pages

2.      Long legends (more than 10 lines each) after each figure disturb the reading of the article and should be included in the article text.

3.      The pathogenesis paragraph is at the same time repetitive and difficult to comprehend; more emphasis with growth factors alteration should be given as an introduction to the following paragraph where the clinical experiences with EGF, PDGF-BB and other tissue-based products are presented.

The paragraph about the new therapies is a rather short text (half a page) followed by a 3 pages table; a more balanced presentation would be welcomed. Moreover, Vac-therapy is still an interesting approach for diabetic foot ulcers but here just two old paper (2005 and 2008) are cited in the table without any comment in the text, while Vac-therapy is still used and studied (see https://doi.org/10.1016/j.jtv.2019.04.001 by Wynn M and Freeman S 2019).

Author Response

Please, kindly find the attchment

Reviewer 3 Report

In this review article, the authors focus on recurrence of diabetic ulcers. Although they well reviewed the literatures of diabetes mellitus, most of the literatures had nothing to do with recurrence of diabetic ulcers. Recurrence of diabetic ulcers is a clinical outcome caused by complicated impairments that underlie diabetes mellitus. It is difficult to relate one factor to recurrence.

The authors focused on recurrence of diabetic ulcers and reviewed the previous articles. However, most of the articles cited (reviewed) are not related to recurrence of diabetic ulcers.

It is interesting for the authors to focus on recurrence of diabetic ulcers. However, the authors’ work (review) did not answer the addressed question. 

No additional information (the subject area compared with other published material) can be found in the manuscript.

Recurrence of diabetic ulcers is a clinical outcome caused by complicated impairments that underlie diabetes mellitus. It is difficult to relate one factor to recurrence.

The authors’ conclusions are just speculations, not based on the evidence.

Most of the articles cited are not related to recurrence of diabetic ulcers.

The figures are just illustrations of the basic impairments that underlie diabetes mellitus. They do not help understand the main question (recurrence of diabetic ulcers).

In conclusion, I do not recommend publication of the submitted article.

Author Response

Reviewer 3.

Dear Sir.

Thank you very much for your excellent comments and constructive criticism. We have attempted to respond as much as we could each and every one of the points. Please, kindly find below our comments side-by-side to yours.

Thank you once again. 

  • What is the main question addressed by the research? The authors focused on recurrence of diabetic ulcers and reviewed the previous articles. However, most of the articles cited (reviewed) are not related to recurrence of diabetic ulcers. – WE AGREE WITH REVIEWER 3. THIS MANUSCRIPT IS NOT FOCUSSING ON THE WELL-REKNOWN CLINICAL ASPECTS OF RECURRENCE; WE RATHER TRIED TO SPOT OVER THE MOLECULAR ENDOGENOUS, UNDISCLOSED, CIRCULATING OR IN-SCAR TISSUE ANCHORED PRIMERS THAT MAY IN THEORY CONTRIBUTE TO REULCERATION. ABOUT THIS NOTHING IS PUBLISHED AS YET. YES, THIS IS A HYPOTHETICAL ANALYSIS AND IS PLETHORIC ON ASSUMPTIONS. • Do you consider the topic original or relevant in the field? Does it address a specific gap in the field? YES SIR. WE DEEM IT IS IMPORRTANT NOT TO FILL THE GAP, BUT SHED AT LEAST HYPOTHETICAL LIGHTS ON THE MANGNITUDE OF THE INTERNAL FACTORS. AS A MATTER OF FACT, IT SEEMS THAT BY ALL MEANS THERE IS NO GOOD CONTROL OF GLYCEMIA DURING THE REMSSION TIME. It is interesting for the authors to focus on recurrence of diabetic ulcers. However, the authors’ work (review) did not answer the addressed question. WE COULD NOT ASPIRE TO ANSWER A QUESTION SIR. NOBODY HAS DONE IT YET. THIS IS AN EMPTY NICHE AT THE INVESTIGATIONAL LEVEL. NO RECURRENCE PREDICITOR HAS EVER BEEN IDENTIFIED. • What does it add to the subject area compared with other published material? SIR, WE DEEM THAT WHAT IT ADDS IS THAT IT IS THE FIRST WORK IN WHICH RECURRENCE IS AUTOPSIED FROM A MOLECULAR PRESPECTIVE AND NOT CIRCUMSCRIBED ON THE WELL KNOWN CLAINICAL AND ORDINARY ASPECTS OF DIET, TYPE OF SHOES, PLANTAR PRESSURE, TOXIC HABITS, ETC, ETC. THIS IS A CAREFULLY STRUCTURED LINE OF THOUGHTS LEADING TO INDICATE THAT THE GHOST OF RECURRENCE LIES ON THE HEART OF SCAR CELLS EPIGENETICS. No additional information can be found in the manuscript. • What specific improvements should the authors consider regarding the methodology? What further controls should be considered? SIR, PLEASE NOTE THAT THIS IS NOT AN EXPERIMENTAL STUDY. IT IS NEITHER A CLINICAL TRIAL. Recurrence of diabetic ulcers is a clinical outcome caused by complicated impairments that underlie diabetes mellitus. It is difficult to relate one factor to recurrence. YOU ARE CORRECT SIR, THERE IS NO FACTOR IDENTIFIED RELATED TO RECURRENCE AND FAR LESS AT THE MOLECULAR LEVEL. THIS IS TURN IS THE MERIT OF THE MANUSCRIPT. WE FILTERED THE LITERATURE THAT COULD BE USEFUL TO MEAKE UP A HYPOTHETICAL ANALYSIS. • Are the conclusions consistent with the evidence and arguments presented and do they address the main question posed? No. The authors’ conclusions are just speculations, not based on the evidence. AGAIN SIR, AND WITH THE MAJOR RESPECT, THIS IS NOT AN EXPERIMENTAL /ORIGINAL RESEARCH. YES, THIS IS A HYPOTHETICAL REVIEW. • Are the references appropriate? No. Most of the articles cited are not related to recurrence of diabetic ulcers. • YOU ARE CORRECT SIR – AS MENTIONED ABOVE, WE DID NOT ADRESS THE WELL-KNOWN RISK FACTORS OF RECURRENCE. THIS EXPLAINS WHY YOU MEAN THAT ARTICLES ARE NOT RELATED TO RECURRENCE. HOWVER, SHOULD I REMARK – OUR CORNERSTONE WAS THE ARMSTRONG, BOULTON AND SISCO’S PAPER PBLISHED ON NEJM IN 2017. THIS IS A SUPER REVIEW OF CLINICAL ASPECTS. Please include any additional comments on the tables and figures. The figures are just illustrations of the basic impairments that underlie diabetes mellitus. ALL THE FIGURE LEGENDS WERE REWORDED. They do not help understand the main question (recurrence of diabetic ulcers). In conclusion, I do not recommend publication of the submitted article

Round 2

Reviewer 2 Report

Excellent work of re-editing has been done!

When a major revision of a paper is asked, some authors prefer to look for another journal and have the paper published with less effort. These authors have accepted all our suggestions, they have done additional efforts, but now can enjoy the result.

Please, highlight the authors name, to request other contributions for MDPI journals in the future!

Reviewer 3 Report

The authors responded well to the reviewers' comments.

The manuscript has now improved after revision.